# Validation of a Rapid Multiresidue Method for the Determination of Pesticide Residues in Vine Leaves. Comparison of the Results According to the Different Conservation Methods

**DOI:** 10.3390/molecules26041176

**Published:** 2021-02-22

**Authors:** Salem Hayar, Rawan Zeitoun, Britt Marianna Maestroni

**Affiliations:** 1Doctoral School of Science and Technology, Research Platform for Environmental Sciences (EDST-PRASE), Lebanese University, Rafic Hariri Campus, Hadath-Mount Lebanon, Lebanon; 2Department of Plant Protection, Faculty of Agricultural Engineering and Veterinary Medicine, Lebanese University, Dekweneh-Beirut, Lebanon; 3Environmental Health Research Lab (EHRL), Faculty of Sciences, Section V, Lebanese University, Nabatieh, Lebanon; rawan.zeitoun@gmail.com; 4Department of Chemistry and Biochemistry, Faculty of Sciences, Section V, Lebanese University, Nabatieh, Lebanon; 5Food and Environmental Protection Laboratory, Joint FAO/IAEA Division of Nuclear Applications in Food and Agriculture, Department of Nuclear Sciences and Applications, International Atomic Energy Agency, Wagramerstrasse 5, A-1400 Vienna, Austria; B.M.Maestroni@iaea.org

**Keywords:** validation, pesticide residues, vine leaves, preservation methods, MRL, QuEChERS, LC-MS/MS

## Abstract

The QuEChERS method was applied to the determination of pesticide residues in vine (*Vitis vinifera*) leaves by LC-MSMS. The method was validated in-house for 33 pesticides representing 17 different chemical groups, that are most commonly used in grape production. Recoveries for the pesticides tested ranged from 75 to 104%, and repeatability and reproducibility relative standard deviations (RSD_r_% and RSD_Rw_%) were less than 20%. The method was applied to the analysis of pesticide residues in 17 market brands of vine leaves processed according to three different preservation methods and sampled from the Lebanese market. Dried vine leaves were more contaminated with pesticide residues than those preserved in brine or stuffed vine leaves. The systemic fungicides were the most frequently detected among all the phytosanitary compounds usually applied to grape production. Brine-preserved and stuffed vine leaves contained lower concentrations of the residues but still contained a cocktail of different pesticides.

## 1. Introduction

Viticulture is one of the major horticultural industries of the world, with the area of cultivated grapevines exceeding 7.5 million ha over a wide range of climate conditions [1,2]. In Lebanon, the cultivated area of grapevines is estimated to be about 10,609 ha, ranking 8th for agricultural productivity in the country [3,4,5]. Viticulture takes advantage of the favourable Mediterranean climatic conditions, with vines growing mostly in Bekaa, Akkar, Mount Lebanon, North and South Lebanon. Vine products are considered valuable agricultural commodities for internal consumption and export, including table grapes, raisins, wine, arrack, vinegar and vine leaves.

Vine leaves are mostly used for cooking [6,7], especially in traditional Mediterranean culinary practices, and are found in the diet of many countries, such as Saudi-Arabia, Lebanon, Cyprus, Turkey, Greece, Spain, Morocco, Italy, Bulgaria and Vietnam [1]. In general, different preservation methods are applied to fresh vine leaves to avoid spoilage, mainly “drying”, “brining” and “bleaching” [7]. In Lebanon, a plate of stuffed vine leaves is one of the famous dishes in Lebanese mezze (appetizers).

Together with grape skins and seeds, vine leaves are considered healthy food products [8,9,10] with homeostatic and astringent properties [11,12,13], antioxidant capacity [14], and anti-microbial and anti-inflammatory activities [15,16,17], due to their high flavonoid and phenol contents [18,19,20]. Vine leaves were also utilized in older traditional medicine [6], where infusions were used to treat hepatitis, haemorrhages, stomach aches and diarrhea, and plant extract preparations were used to heal abscesses and wounds [21].

Recently, several pharmaceutical companies have made large investments in an attempt to find an effective and safe source of phenols [22,23]. The phenolic and other non-phenolic compounds in various grapevine parts such as berries, stems, petioles, leaves, and shoots have been of great research interest [24]. Investigations with the Lebanese private industry highlighted that the annual income from exporting vine leaves to the Gulf countries can reach up to two million US dollars (personal communication). In general, there is a growing demand for vine leaves in international food markets [1].

Grapevines, like any other plant species, are exposed to environmental influences, and threatened by several pests and diseases, including fungal diseases such as downy mildew (*Plasmopora viticola*), powdery mildew (*Uncinula necator*) and gray mold (*Botritis cinerea*), viral diseases such as feanleaf virus (GFLV) and leaf roll (GLRaV-1, GLRaV-2), phytoplasma diseases including flavescence dorée (FD), insect pests such as grape moth (*Lobesia botrana*), and assorted aphids, thrips, beetles and mites [25]. Therefore, a wide range of pesticides are usually applied to vineyards at various stages of cultivation and during post-harvest storage, to prevent deterioration of vine crops and to control pests and plant pathogens. Winegrowers mostly use fungicides, and sometimes also insecticides, acaricides, and herbicides [2,26]. In Lebanon, in the frame of vineyard protection, the most common pesticides used belong to the chemical groups; triazoles, benzimidazoles, strobilurins, anilinopyrimidines, pyrethroids, and organophosphorous, neonicotinoid and carboximides compounds, each used for a different disease [4,5]. Often phytosanitary treatments are preventative rather than curative, and in some cases, good agricultural practices are lacking. For example, the scheduled pesticide applications may be composed of a cocktail of pesticides with inappropriate selection of chemicals, with the potential for misuse and overuse, and the crops may be harvested before the pre-harvest interval indicated on the label of the formulations, which may lead to serious and detrimental consequences on the ecosystems and impact human health. Therefore, not only the fruit, as a target crop, but also the leaves, as a secondary crop, are exposed to significant contamination by pesticides.

To control the legitimate use of pesticides, international organizations around the world set levels for the concentrations of pesticide residues that are legally allowed in food products as they enter the market. These levels of agrochemicals are commonly referred to as Maximum Residue Levels or MRLs [27,28] and are set for specific commodities and compounds taking into consideration factors such as the quantity and frequency of use of the pesticide on a crop, experimental data on expected residues when the pesticide is applied according to GAP, the toxicological reference values for the pesticides and the acceptable daily intake (ADI) and acute reference dose (ARfD) values. If a safe MRL cannot be recommended, or when an MRL has not been set for a particular commodity and compound, the default MRL is usually established by the EU at an estimated lower limit of analytical determination: “*A general default MRL of 0.01 mg/kg applies where a pesticide is not specifically mentioned*” [29]. Indeed, MRLs are the highest levels of residues expected to be in the food product when the pesticide is used according to authorised agricultural practices [30,31]. The MRLs are always set far below levels considered to be harmful for humans. Grapevine leaves are a side product of grape production, with grapes being considered the main target crop; vineyards have never been cultivated just for the sake of producing vine leaves. As a result, very few studies have been conducted to evaluate the maximum safe levels for pesticides on vine leaves.

A study done in Turkey identified 42 different pesticides in 36.6% of the tested vine leaves and 22.4% of those samples contained pesticide residues at levels above the MRLs; metalaxyl and azoxystrobin were the common detected pesticides [32].

The systemic fungicides carbendazim, cyproconazole, tebuconazole, penconazole and two contact insecticides chlorpyrifos and lambda-cyhalothrin were the most frequently detected pesticides, being found in more than 50% of tested vine leaves [33,34].

No local or internationally-recognised risk assessment bodies, such as the European Food Safety Authority (EFSA) or the Joint Meeting on Pesticide Residues (JMPR) for Codex Alimentarius, have specified MRLs or performed risk assessment exercises for pesticide residue levels in grapevine leaves. The lack of an MRL and the consequent errors in interpretation of requirements constitute a major trade barrier, limiting the export of Lebanese grapevine leaves and leading to missed economic profits and financial losses.

In recent years a very limited number of compounds have been studied in vine leaves, for example, trifloxystrobin, tebuconazole [35], fipronil [36], imidacloprid [37], azoxystrobin, fenhexamid and lufenuron [38]. Only two studies are available for the multi residue validation in grape vine leaves. Authors reported a gas chromatography coupled to tandem mass spectrometry GC-MS/MS method where the sample preparation involved a modified and miniaturized SweEt/QuEChERS method, which uses acidified ethyl acetate for extraction (SweEt) and cleanup using a modified QuEChERS procedure. This method was only tested for 59 GC-amenable pesticides [39]. Other authors developed a comprehensive analytical method for the identification and quantification of a broad range of pesticides and plant growth regulators in vine leaves matrix using an optimized QuEChERS-based extraction protocol and determination by GC-MS/MS and liquid chromatography coupled to tandem mass spectrometry LC-MS/MS. Both methods were validated showing satisfactory recoveries, accuracy, precision in compliance with CODEX and SANTE’s method performance criteria [30,40].

The current study aims to: (i) establish a validated method for pesticide residue analysis in vine leaves by liquid chromatography coupled to tandem mass spectrometry (LC-MS/MS) and adapt it to the local laboratory conditions, (ii) assess the current level of pesticide residues in *V. vinifera* leaves on the Lebanese market, (iii) compare the pesticide residue content in vine leaves processed using three different preservation methods. The ultimate goal is to contribute to the protection of consumer health and to the enhancement of the regional trade of vine leaves.

## 2. Results and Discussion

### 2.1. Method Validation

In the validation experiments, before spiking the analytical portions with the required amount of pesticide mixtures, the blank samples were tested and checked for the absence of any of the target pesticides.

The validation of the method was performed using the 33 compounds, representing 17 classes of pesticides that are considered to be the most commonly used pesticides for grape production in Lebanon [5]. These pesticides are listed in Appendix A.

The method performance criteria achieved are reported in Table 1.

The mean recoveries (RM%) over the analytical range varied between 75 and 103% as shown in Table 1. The recovery values were in line with those reported for the analytes commonly validated [39,41].

Linearity was achieved for all pesticides, with coefficients of regression (R^2^) better than 0.99.

Method accuracy and precision were checked by the determination of the within laboratory repeatability (RSD_r_%) and reproducibility (RSD_Rw_%) of the recovery results (Table 1). Both RSD_r_% and RSD_Rw_% were less than 20% in all cases, which is in accordance with the guidelines (EU SANTE/12682/2019) [40].

The limit of quantification (LOQ) and the limit of detection (LOD) were lower than the corresponding default EU-MRLs for vine leaves, rendering the method acceptable for checking compliance to MRLs. The values are listed in Table 1.

The LOD of the method was between 0.84 and 2.43 ng/g and the LOQ was below 8.02 ng/g for all compounds. The LOQs were lower than the corresponding EU-MRLs for vine leaves, rendering the method acceptable for checking compliance to MRLs.

The analyte stability during the washing procedure of the market samples was also checked as part of the method validation. Previous studies showed that the washing procedure with deionized water did not significantly affect the residues in the stuffed vine leaves, even for the water-soluble compounds such as acetamiprid, dimethoate and metalaxyl comparing to lipophilic molecules as chlorpyrifos and difenoconazole (unpublished data). Indeed, the results shown in this study present an inventory of the levels of pesticide residues that may occur in the 3 methods of preservation to point out which of these preservation methods is safer and healthier for the consumer.

The method performance was in compliance with the analytical quality control criteria of the EU SANTE/12682/2019 guideline and therefore considered fit for purpose [40].

The method was therefore used in the national residue monitoring programme for pesticide residues in vine leaves.

### 2.2. Assessment of the Actual State of the Lebanese Market

Preserved grapevine leaves, from brands that are commonly found in the Lebanese market, were analysed to establish the types and the concentrations of pesticide residues present in order to assess the actual situation. Twenty-four different samples of dry, brine and stuffed preserved leaves were randomly collected from various points of sale and analysed. Results are presented in Table 2.

In the present investigation, 33 pesticide molecules, that belong to 17 different chemical classes, were detected and quantified. Systemic fungicides and insecticides were the most commonly detected pesticides, followed by three contact acaricides. Carbendazim, chlorpyrifos, lambda-cyhalothrin, fenazaquin, pyraclostrobin, cyproconazole and tebuconazole were the most common compounds found among the various samples and mostly in the samples of the dry preserved leaves (Figure 1). These phytosanitary products were also the most commonly detected during a monitoring survey of 588 samples collected from all Lebanese vineyard regions [5]. However, imazalil, imidacloprid, diazinon, deltamethrin, pyridaben and cyprodinil, which are commonly applied in grape protection (Figure 1) were not detected in the samples regardless of the means of preservation.

#### 2.2.1. Dry preserved Samples

As shown in Table 2, the dry preserved samples generally contained a very high number of pesticide residues per sample and also the highest concentrations of the residues (Table 2). The brands “A”, “B”, “D”, and “L” contained a cocktail of 10 to 14 pesticides with more than the half containing concentrations exceeding the corresponding MRL (Figure 2). Fenazaquin (acaricide/insecticide) was the most frequently detected pesticide, followed by carbendazim, chlorpyriphos, lambda-cyhalothrin and cyproconazole, which are present in more than 60% of the dry vine leaves samples tested. As for levels of residues, the two fungicides kresoxim-methyl and boscalid exceeded the default EU MRL (0.01 mg/kg) by 415 and 277 times, respectively.

Similar results were reported by the European Rapid Alert System for Food and Feed (RASSF) in 2016 and 2020 [42], where different shipments of pickled vine leaves from Turkey and Egypt were rejected at the borders of Bulgaria and Germany, with the samples containing a cocktail of more than 16 pesticide molecules (notification 2016.ABR & 2020.0227) [42].

#### 2.2.2. Brine Preserved Samples

In samples preserved in brine, mixtures of up to 14 active substances were found for the market samples “F”, “G”, “M” and “N” (Figure 2). The concentrations of pesticide residues were lower than those found in dry preserved samples. The two systemic fungicides, difenoconazole and boscalid, were detected at the highest levels in the samples “N” and “X” at concentrations of 4.92 and 3.54 mg/kg respectively, followed by the systemic insecticide/acaricide lufenuron (2.84 mg/kg) in sample “M”. This is in line with the Greek notification in 2003 that an unauthorized systemic fungicide, procymidone, was detected in vine leaves preserved in brine in a shipment from Turkey. The concentration found was 22.4 mg/kg, which is 224 times higher than the EU MRL (notification 2003.AYW) [42].

#### 2.2.3. Stuffed Preserved Samples

As shown in Figure 2, stuffed vine leaves can also contain a cocktail of pesticide residues; samples “I”, “H”, “J” and “S” having been found to contain 12, 9, 8 and 6 different molecules in the same sample, respectively. The concentrations of the active substances were lower than the other modes of vine leaves preservation. Only the systemic fungicide, boscalid, reached a concentration of 1.05 mg/kg in a single brand sample, “I”. A possible explanation for the lower levels found is that the pesticide levels are reduced by the pasteurization process that is used in the preparation of stuffed vine leaves Indeed, of the 48 notifications declared by the RASSF over the past 20 years (2000–2020) preservation, only 2 cases were for stuffed vine leaves, each containing a single active substance; the non-systemic insecticide chlorpyrifos (2.8 mg/kg) and the non-systemic acaricide propargite (0.2 mg/kg) (notifications 2013.CCA and 2016.AAC) [42].

#### 2.2.4. Comparison of the Processing Modes

The results show that among the three modes of preservation, the pesticide residue concentrations in the dry vine leaves were generally found to be very high compared to the brine-preserved and stuffed leaves. This is in agreement with a previously reported study [32].

The cooking process seems to reduce the pesticide residue concentrations, though without decreasing the number of pesticides detectable in the sample. Similar findings were revealed by [34]. They observed that application of the hot brine to vine leaves is effective in reducing pesticide residue concentrations in vine leaves of the Sultani seedless grape variety, which is in agreement with the findings in the current study (Table 2). In the 48 RASSF notifications from 2000 to 2020, it was mainly the same pesticide residues that were detected. Several authors have reported that food processing causes a decrease in pesticide content [43,44,45,46]. Cooking was more effective than washing for the removal of chlorpyrifos residue from five types of vegetables (cabbage, garlic sprouts, tomato, cucumber, eggplant) [47]. Washing with hot water was very effective in reducing residues of the polar pesticide, dimethoate, but did not have a significant effect on residues of the apolar compound, chlorpyrifos [48]. Morever, deltamethrin, permethrin, cypermethrin and chlorpyrifos were reduced to 59.9–86.4% and 63.2–90.2% during dough preparation and baking, respectively [49]. A total reduction of 87.98%, 73.69%, 85.93%, 71.31%, 78.18%, and 90.33% for deltamethrin, penconazole, kresoxim-methyl, cyproconazole, epoxiconazole and azoxystrobin were found respectively in a study on the effect of household rice cooking [46].

Other literature has revealed that pesticide residue values in the leaves stored without brine (dry) were found to be very high compared to the brine medium at two different temperatures. The pesticide residue values decrease in cold brine by 69–73% and by 73–91% in hot brine compared to vine leaves without brine (dry) [34].

It is important to note that no traceability information was available for the plant materials of the tested samples or the place of origin and date of harvesting of the vine leaves for the market samples. In addition, it was difficult to make inquiries with farmers to collect information about the agricultural practices, such as the type, dose and frequency of pesticide formulations applied to the vineyards. In the absence of targeted legislation, it was difficult to evaluate if the preharvest intervals were implemented.

It was notable that the iMethod application of the LC-MS/MS system allowed the detection, in screening mode, of two new fungicides namely cyflufenamid and fludioxonil, that belong to two chemical classes (the phenylpyrroles and amidoxines, respectively) that are not currently registered for application on grapes in Lebanon. This finding suggests that in some cases the raw material used in conserved vine leaves may be imported from other countries.

## 3. Materials and Methods

### 3.1. Samples of Vine Leaves

#### 3.1.1. Market Samples

Twenty-four different samples of vine leaves of 17 well-known trademarks were collected randomly from various points of sale throughout Lebanon. The commercial samples were stored in the dark at room temperature (22 °C) until analysis.

#### 3.1.2. Blank Matrix Sample

Three batches of 750 g of *Vitis vinifera* leaves were harvested, during early grapevine growing season, from vineyards located in Bekaa, at an altitude of 900 m above sea level, in moderate climate conditions. The leaves did not undergo any phytosanitary treatment before their harvest and were used as a blank matrix in this study.

### 3.2. Chemicals, Materials and Standards

Organic solvents such as acetonitrile and methanol were of analytical grade and purchased from Sigma-Aldrich Chemie GmbH (Munich, Schnelldorf, Germany). Purified water was prepared using a Milli-Q water purification system (Millipore, Billerica, MA, USA). QuEChERS materials (NaCl, anhydrous MgSO_4_, PSA and GCB) were purchased from Agilent Technologies (Santa Clara, CA, USA). Pesticide standards (purity ranging between 96 and 99.5%) were purchased from Dr. Ehrenstorfer (Munich, Germany) and Sigma-Aldrich Chemie GmbH.

### 3.3. Standard Solution Preparation

A mixture of 33 certified pesticide analytical standards was used for the quantification of pesticide residues. The individual stock solution of each molecule (10 mg/mL) was prepared in acetonitrile and stored at −18 °C. The mixed stock solution was prepared in acetonitrile at 25 ng/µL and working standard solutions were prepared by serial dilutions with six levels of concentrations from 0.01–0.1 µg/g; (5–500 µg/L). All working solutions were stored in the dark at 4 °C. The pesticides included in the mixture are presented in Table 1.

### 3.4. Sample Preparation

The extraction of pesticide residues was performed according to the QuEChERS sample preparation protocol [50]. Regarding the stuffed vine leaves brands purchased from the Lebanese market, the vine leaves were separated from the other ingredients and gently washed with deionized water and dried with laboratory tissue paper; then the leaves were subjected to the same extraction protocol as described below.

The market samples came as glass jars filled with vine leaves. The full content of the glass jars was finely homogenized using a VCM4 Waring Vertical Cutter Blender/Mixer (Hallde, Sweden) according to the instructions described in the U.S. Food and Drug Administration’s Pesticide Analytical Manual [51]. The comminution of the 950 g vine leaves at maximum speed for 5 min produced a homogeneous laboratory sample, from which analytical portions of 10 g were analysed.

10 g of homogenized vine leaves were accurately weighed and transferred to a polypropylene centrifuge tube with screw cap. Ten mL of acetonitrile, 4 g of MgSO4 and 1 g of NaCl were added to the tube and the mixture was shaken vigorously for 1 min, and then centrifuged for 10 min at 2066× *g*. An aliquot of 1 mL of the acetonitrile phase was transferred into a 15-mL dispersive solid-phase extraction (d-SPE) tube containing 150 mg MgSO_4_ to remove the water from the organic phase, 25 mg primary secondary amine (PSA) to remove various polar organic acids, polar pigments, some sugars and fatty acids; and 50 mg graphitised carbon black (GCB) to remove sterols and pigments. Then, the tube was closed and shaken vigorously for 1 min and centrifuged for 10 min at 3000 rpm. The extract was isolated immediately, put in a new 15-mL polypropylene centrifuge tube and left in a refrigerator overnight, then filtered through a 0.20 μm PTFE filter and transferred into a glass vial to be analysed by LC-MS/MS. When needed, the final sample extracts were diluted with acetonitrile to fall within the quantification range given by the calibration curves.

### 3.5. LC-MS/MS Analysis

The quantification of pesticide residues was performed by LC-MS/MS. The instrument used was a 1200 Infinity series liquid chromatography system (Agilent Technologies) coupled to tandem mass spectrometer. Chromatographic separation was performed with a reverse-phase analytical C_18_ column of 150 × 2 mm and 2.5 μm particle size, Synergi (Phenomenex, Torrance, CA, USA) equipped with a guard-column. The mobile phase consisted of a water-methanol solvent containing each one 5 mM ammonium acetate. The gradient program was as follows: 2% B to 100% of B over 12 min, held at 100% B until 20 min then decreased to 2% B at 25.01 min. The total run time was 30 min with a flow rate of 0.4 mL min^−1^. The injection volume was 5 μL and the column was maintained at 25 °C. The LC was coupled to a 3200 QTRAP Triple Quadrupole Mass Spectrometer (AB SCIEX, Dublin, CA, USA) fitted with an electrospray chemical ionization (ESI) source and operated in positive ion mode. Data acquisition was performed in multiple reaction monitoring (MRM) mode. The ion spray voltage was 5 kV and the source temperature were set at 500 °C. The collision energy (CE), the declustering potential (DP), the entrance potential (EP) and the collision cell exit potential (CXP) were optimised for each target analyte-pesticide (Appendix A). Nitrogen gas was used as a collision gas and nebulizer curtain gas.

The system was equipped with a pre-configured iMethod™ Application (AB Sciex) and associated libraries. This was designed for either quantitative analysis or for qualitative screening using QTRAP^®^ technology and therefore can be used for the routine screening identification of up to 400 pesticides. Figure 3 shows the Total Ion Chromatogram (TIC) of the MRMs for the pesticide mixture at 100 µg/L and the Extracted Ion Chromatogram for the difenoconazole in a vine leaves sample.

### 3.6. Identification and Quantification of the Pesticide Residue

The identity of a pesticide residue in an extract was confirmed by its retention time matched with that of the appropriate analyte in the pure standard solutions and the appearance of two product ion transitions that matched the relative intensity criteria specified by the EU SANTE/12682/2019 guideline [40]. Once the presence of a pesticide residue was confirmed in an extract, the concentration of the residue was obtained from the appropriate calibration function.

### 3.7. Method Validation Criteria

The method was applied to the analysis of five replicate analytical portions (10 g) spiked at 0.01; 0.05; and 0.1 μg/g with the mixture of pesticides and the experiment was repeated on 3 different days.

The verification of the method performance criteria was done according to the EU SANTE/12682/2019 guidelines only for the following parameters: instrument linearity, method recovery, repeatability, reproducibility, and limit of quantification (LOQ) [40]. The limit of detection (LOD), although not a requirement of the 2019 version of the SANTE guideline, was calculated to be able to compare method performance criteria with other methods published in the literature. The limits of detection (LODs) and quantification (LOQs) were estimated following the IUPAC approach and SANCO/12495/2011 guideline [52] which consisted of analysing the blank sample to establish noise levels and then testing experimentally estimated LODs and LOQs for signal/noise, with target ratios of 3 and 10, respectively [53].

The linearity was checked by the back-calculated concentration as explained in the guidelines, and in addition by the visual verification of the linearity of the calibration curve prepared from six concentration levels for each analyte (ranging from 5 to 500 µg/L) in solvent. Within-laboratory repeatability and reproducibility were checked by measuring the average recoveries at 3 different concentration levels (0.01; 0.05; and 0.1 μg/g) with five replicates and on 3 different days.

## 4. Conclusions

A precise, accurate, and reliable method for determination of pesticide residues in vine leaves using LC-MS/MS has been developed, validated and applied. The values obtained for recoveries (RM%) over the analytical range varied between 75 to 104%, with repeatability (RSD_r_) 4–18% and reproducibility (RSD_Rw_) 4–19%, revealed that the use of the method in the laboratory provides good linearity, accuracy, and precision for all analytes. Hence, it is suitable and convenient for the routine determination of pesticide residues in vine leaves samples. The method was successfully applied to the determination of pesticide residues in 24 samples from 17 brands of vine leaves, with three different modes of preservation, available on the Lebanese market. In general, the results shown that the dry preserved vine leaves contained the highest levels of pesticide residues compared to brine-preserved and stuffed leaves. Furthermore, the systemic pesticide residues were more frequently found than the contact pesticides. A cocktail of pesticide residues containing up to 13 molecules were detected in the same sample, some at concentrations far exceeding the default MRLs (0.01 mg/kg) set in EU legislation. The systemic fungicides carbendazim, cyproconazole and tebuconazole were more frequently detected than the non-systemic insecticides chlorpyrifos, lambda-cyhalothrin and fenazaquin.

Despite the fact that the Lebanese Ministry of Agriculture (MoA) recently banned 13 of the 33 pesticides detected in vine leaves; the MoA must continue its efforts by conducting several extension training programs for vineyard growers on grape leaves production. Field guides and booklets on an appropriate pesticide application control program for the production of both grapes and vine leaves must be developed and made available to farmers to enable them to: (i) recognize the pests and crop diseases, (ii) ensure the proper use of pesticides and avoid the mixture of different types of formulations and (iii) reduce the pesticide residue problem on vine leaves during the growing period.

Further investigations are needed to understand the effect of the 3 modes of preservation (dry, brine and bleaching) on the behavior and dissipation of pesticide residues in vine leaves during storage.

## Figures and Tables

**Figure 1 molecules-26-01176-f001:**
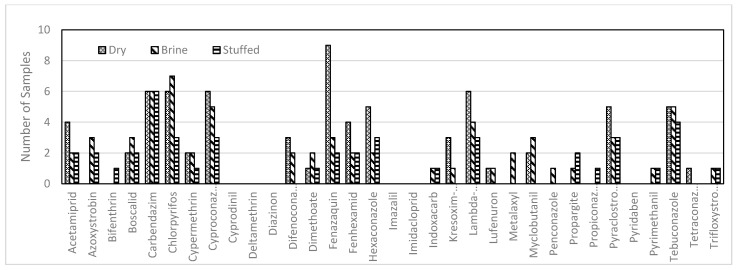
Number of samples containing the active substance and its distribution between the 3 preservation methods.

**Figure 2 molecules-26-01176-f002:**
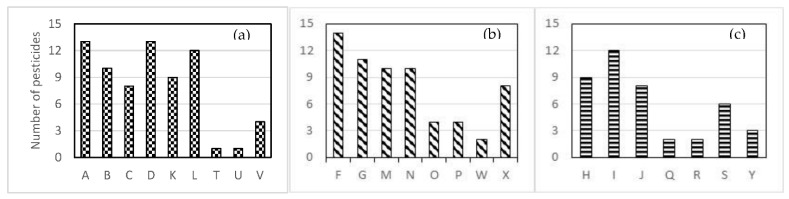
Cocktails of pesticide residues in dry (**a**), brine-preserved (**b**) and stuffed vine leaves (**c**) albeit at the lowest concentration levels.

**Figure 3 molecules-26-01176-f003:**
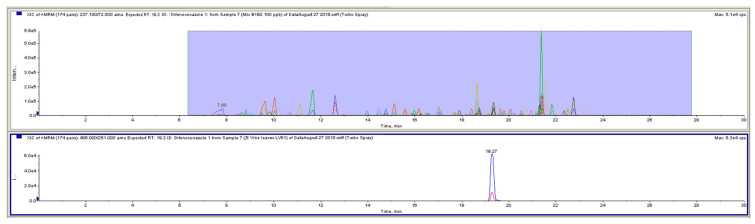
Total Ion Chromatogram (TIC) of the MRMs for the pesticide mixture at 100 µg/L (upper figure) and the Extracted Ion Chromatogram for the difenoconazole in a vine leaves sample (lower figure).

**Table 1 molecules-26-01176-t001:** Average of recovery data (RM%), repeatability (RSD_r_%) and reproducibility (RSD_Rw_%) for the 33 pesticides at the three fortification levels, 0.01, 0.05 and 0.1 µg/g (n = 5 at each level).

			Level Spiking (0.01 µg/g)			Level Spiking (0.05 µg/g)			Level Spiking (0.1 µg/g)		
Pesticide	LOD	LOQ	RM	RSD_r_	RSD_Rw_	RM	RSD_r_	RSD_Rw_	RM	RSD_r_	RSD_Rw_
	(µg/g)		(%)			(%)			(%)		
Acetamiprid	0.002	0.006	99	6	6	101	5	4	96	6	7
Azoxystrobin	0.002	0.006	85	12	13	84	7	8	82	8	7
Bifenthrin	0.001	0.005	97	9	12	97	4	8	97	5	7
Boscalid	0.002	0.007	90	8	8	87	7	7	84	9	12
Carbendazim	0.001	0.005	91	8	12	87	12	11	88	7	8
Chlorpyriphos	0.001	0.004	82	11	12	95	7	10	94	9	11
Cypermethrin	0.002	0.006	88	10	15	87	11	10	89	8	12
Cyproconazole	0.001	0.003	78	11	12	80	12	14	80	12	13
Cyprodinil	0.002	0.008	98	10	13	86	8	10	86	8	12
Deltamethrin	0.001	0.004	86	10	12	76	7	8	80	4	7
Diazinon	0.001	0.003	82	14	12	85	7	5	85	5	5
Difenoconazole	0.002	0.006	96	14	12	98	11	14	94	12	15
Dimethoate	0.002	0.007	87	9	7	83	4	5	82	6	10
Fenazaquin	0.002	0.007	101	12	8	104	15	16	100	12	12
Fenhexamid	0.001	0.004	84	8	7	80	9	7	77	7	8
Hexaconazole	0.002	0.006	98	17	15	100	12	14	94	14	16
Imazalil	0.002	0.008	88	17	14	90	14	10	91	12	12
Imidacloprid	0.002	0.006	92	17	19	84	7	8	82	11	13
Indoxacarb	0.001	0.004	83	17	19	84	13	12	80	14	12
Kresoxim-methyl	0.002	0.006	83	10	11	87	9	11	83	8	12
Lambda-Cyhalothrin	0.001	0.004	78	14	13	75	8	12	76	8	10
Lufenuron	0.002	0.006	85	11	14	83	9	12	82	10	13
Metalaxyl	0.002	0.006	87	18	16	83	12	14	82	13	12
Myclobutanil	0.001	0.004	98	8	10	97	4	7	95	7	6
Penconazole	0.002	0.007	96	12	14	98	10	13	96	14	15
Propargite	0.001	0.004	87	8	7	89	8	9	88	7	8
Propiconazole	0.001	0.005	88	11	14	87	12	12	88	11	12
Pyraclostrobin	0.002	0.005	75	8	7	75	4	8	78	5	4
Pyridaben	0.001	0.004	103	14	15	98	12	12	97	11	10
Pyrimethanil	0.002	0.005	90	12	12	89	8	7	87	8	7
Tebuconazole	0.002	0.005	94	10	9	95	11	12	92	12	11
Tetraconazole	0.002	0.007	95	12	14	95	10	12	94	10	12
Trifloxystrobin	0.001	0.004	88	10	8	87	7	8	87	7	6

**RM**: Recovery mean, **RSD_r_**: Repeatability, **RSD_Rw_**: Reproducibility (within laboratory).

**Table 2 molecules-26-01176-t002:** Pesticide residue concentrations (mg/kg) for the dry, brine and stuffed preserved grapevine leaves market samples compared to EU-MRLs.

Pesticide	MRL (mg/kg)	Dry Conservation Brands									Brine Conservation Brands								Stuffed grapevine leaves Brands						
		A	B	C	D	K	L	T	U	V	F	G	M	N	O	P	W	X	H	I	J	Q	R	S	Y
Lufenuron	0.02 *	ND	ND	ND	0.1	ND	ND	ND	ND	ND	ND	ND	2.8	ND	ND	ND	ND	ND	ND	ND	ND	ND	ND	ND	ND
Carbendazim	0.1 *	0.07	0.1	0	0.2	0.2	1.13	ND	ND	ND	0.8	0	0	0.4	1.3	1.3	ND	ND	0	0.4	0	0.1	0.1	0	ND
Boscalid	0.05 *	13.9	ND	ND	ND	ND	3.21	ND	ND	ND	1.5	2.4	ND	ND	ND	ND	ND	3.5	ND	1.1	ND	ND	ND	0.2	ND
Acetamiprid	0.01 *	1.48	0	ND	0.1	ND	0.17	ND	ND	ND	ND	ND	ND	0.1	ND	ND	ND	0.7	ND	ND	ND	ND	ND	ND	ND
Imidacloprid	2	ND	ND	ND	ND	ND	ND	ND	ND	ND	ND	ND	ND	ND	ND	ND	ND	ND	ND	ND	ND	ND	ND	ND	ND
Fenhexamid	0.05 *	0.02	0	ND	0	0	ND	ND	ND	ND	ND	0.8	ND	0	ND	ND	ND	ND	0	0	ND	ND	ND	ND	ND
Imazalil	0.05 *	ND	ND	ND	ND	ND	ND	ND	ND	ND	ND	ND	ND	ND	ND	ND	ND	ND	ND	ND	ND	ND	ND	ND	ND
Chlorpyrifos	0.05 *	0.44	0.1	0.1	0.1	0.1	0.05	ND	ND	ND	0.7	0.1	0.6	0.1	0.9	1.2	ND	1.6	0.1	0.1	0.1	ND	ND	ND	ND
Diazinon	0.01 *	ND	ND	ND	ND	ND	ND	ND	ND	ND	ND	ND	ND	ND	ND	ND	ND	ND	ND	ND	ND	ND	ND	ND	ND
Dimethoate	0.02 *	ND	ND	ND	ND	ND	ND	ND	ND	0.2	ND	ND	ND	ND	ND	ND	0.2	0.2	ND	ND	ND	ND	ND	ND	0.2
Indoxacarb	0.02 *	ND	ND	ND	ND	ND	ND	ND	ND	ND	ND	ND	ND	ND	ND	ND	ND	0.9	ND	ND	ND	ND	ND	ND	1
Metalaxyl	0.05 *	ND	ND	ND	ND	ND	ND	ND	ND	ND	0	ND	ND	ND	ND	ND	ND	1.4	ND	ND	ND	ND	ND	ND	ND
Bifenthrin	0.05 *	ND	ND	ND	ND	ND	ND	ND	ND	ND	ND	ND	ND	ND	ND	ND	ND	ND	ND	0	ND	ND	ND	ND	ND
Cypermethrin	0.05 *	ND	ND	ND	0.5	ND	5.62	ND	ND	ND	ND	ND	ND	0.2	ND	1.8	ND	ND	ND	ND	0.2	ND	ND	ND	ND
Deltamethrin	0.5	ND	ND	ND	ND	ND	ND	ND	ND	ND	ND	ND	ND	ND	ND	ND	ND	ND	ND	ND	ND	ND	ND	ND	ND
Lambda-Cyhalothrin	0.02 *	0.08	0.1	1	0.1	0.1	0.06	ND	ND	ND	0.1	0.1	0.1	1.1	ND	ND	ND	ND	0.1	0.1	0.1	ND	ND	ND	ND
Pyridaben	0.05 *	ND	ND	ND	ND	ND	ND	ND	ND	ND	ND	ND	ND	ND	ND	ND	ND	ND	ND	ND	ND	ND	ND	ND	ND
Cyprodinil	0.05 *	ND	ND	ND	ND	ND	ND	ND	ND	ND	ND	ND	ND	ND	ND	ND	ND	ND	ND	ND	ND	ND	ND	ND	ND
Pyrimethanil	0.01 *	ND	ND	ND	ND	ND	ND	ND	ND	ND	0.4	ND	ND	ND	ND	ND	ND	ND	ND	0	ND	ND	ND	ND	ND
Fenazaquin	0.01 *	0.59	0.3	0.4	0.5	0.8	0.5	0.1	0.2	0.3	ND	ND	ND	0.1	0.2	0.2	ND	ND	ND	ND	ND	0.1	0.1	ND	ND
Azoxystrobin	0.05 *	ND	ND	ND	ND	ND	ND	ND	ND	ND	1.1	ND	ND	ND	0.1	ND	ND	0.9	ND	0.2	ND	ND	ND	0	ND
Kresoxim-methyl	0.05 *	9.61	ND	ND	ND	ND	20.7	ND	ND	8.8	0.2	ND	ND	ND	ND	ND	ND	ND	ND	ND	ND	ND	ND	ND	ND
Pyraclostrobin	0.02 *	0.02	0	0	0	0	ND	ND	ND	ND	0	0.2	0	ND	ND	ND	ND	ND	0	0	0	ND	ND	ND	ND
Trifloxystrobin	0.02 *	ND	ND	ND	ND	ND	ND	ND	ND	ND	ND	0	ND	ND	ND	ND	ND	ND	ND	ND	ND	ND	ND	ND	0.1
Propargite	0.01 *	ND	ND	ND	ND	ND	ND	ND	ND	ND	0.7	ND	ND	ND	ND	ND	ND	ND	ND	0.5	ND	ND	ND	0.1	ND
Cyproconazole	0.05 *	0.01	0	0.1	0.1	0	0.01	ND	ND	ND	0	0	0.1	0	ND	ND	0.1	ND	0	0	0	ND	ND	ND	ND
Difenoconazole	0.05 *	0.01	0.2	ND	ND	ND	ND	ND	ND	0.3	ND	ND	0	4.9	ND	ND	ND	ND	ND	ND	ND	ND	ND	ND	ND
Hexaconazole	0.01 *	0.02	ND	0	0.3	0	0.02	ND	ND	ND	ND	0	0.6	ND	ND	ND	ND	ND	0.1	0	0	ND	ND	ND	ND
Myclobutanil	0.02 *	ND	ND	2.4	0	ND	ND	ND	ND	ND	0.1	0.4	1.8	ND	ND	ND	ND	ND	ND	ND	ND	ND	ND	ND	ND
Penconazole	0.05 *	ND	ND	ND	ND	ND	ND	ND	ND	ND	0.1	ND	ND	ND	ND	ND	ND	ND	ND	ND	ND	ND	ND	ND	ND
Propiconazole	0.05 *	ND	ND	ND	ND	ND	ND	ND	ND	ND	ND	ND	ND	ND	ND	ND	ND	ND	0.1	ND	ND	ND	ND	ND	ND
Tebuconazole	0.02 *	0.01	0	ND	0	0	0.01	ND	ND	ND	0.6	0	0	0	ND	ND	ND	0.6	0	0	0	ND	ND	0.1	ND
Tetraconazole	0.02 *	ND	ND	ND	ND	ND	0.03	ND	ND	ND	ND	ND	ND	ND	ND	ND	ND	ND	ND	ND	ND	ND	ND	0.1	ND
**Time (Months) Θ**		**1**	**2**	**1**	**13**	**13**	**8**	**ND**	**14**	**8**	**2**	**12**	**15**	**14**	**3**	**3**	**8**	**9**	**9**	**1**	**7**	**3**	**3**	**5**	**3**
**Nr. pesticide/sample**		**13**	**10**	**8**	**13**	**9**	**12**	**1**	**1**	**4**	**14**	**11**	**10**	**10**	**4**	**4**	**2**	**8**	**9**	**12**	**8**	**2**	**2**	**6**	**3**
**Nr. pesticide >MRL**		**7**	**5**	**5**	**10**	**5**	**9**	**1**	**1**	**4**	**11**	**5**	**6**	**7**	**4**	**4**	**2**	**8**	**3**	**7**	**3**	**1**	**1**	**4**	**3**

Values are the average of triplicate analysis of each sample. MRLs are those imposed by the European Commission (EU Pesticides Database, Reg. No. 396: 2005) on products with code number 0253000: vine leaves (grape leaves). The symbol (*) indicates the lower limit of determination corresponding to the molecule (EU/Reg. No. 34:2013). (ND) indicates that the pesticide is not detected. The symbol (^Θ^) indicates the time (Months) between the production date and the pesticide residues extraction date.

## Data Availability

The data presented in this study are openly available in this article.

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
