# Peer review of "Validation of a Rapid Multiresidue Method for the Determination of Pesticide Residues in Vine Leaves. Comparison of the Results According to the Different Conservation Methods"

_molecules, 2021, doi:10.3390/molecules26041176_

Round 1

Reviewer 1 Report

The study highlight the important concern about deregulated use of pesticides in Labanese marker of vine leave. The authors have validated the method and analyzed 17 marketed vine leaves in 3 forms. The aim of the study is to validate the method, assess the current level of pesticide V. Vinifera leaves on the Lebanese market.

However, the study raises following concerns about method and conclusion.

  1. How the major observation about more pesticides found in dried leave is validated. For example; author should test that sample preparation of brine and stuffed samples does not eliminate any pesticides particularly, the washing step. It might be the case that pesticides are eliminated during sample preparation and washing which gives the impression that dried leaves have more pesticides. 
  2. The stability of pesticides during storage and preparation is not checked. 
  3. The above two points raises some concern about the comparative data presented for 3 types of samples. 
  4. Is the LC-MS method new or adapted from literature. What is the improvement or advantage of the LC-MS method over previous reports. 
  5. Author didn’t explain the observed pesticide levels and their possible causes w.r.t. sample sources. 
  6. Method section need more details about sample collection, properties, selection criteria, LC-MS method optimization, selection of MS/MS transitions, LC column (C18 or C8)
  7. Identification of pesticides is not clear. Please provide some examples of MS/MS spectra from standard and sample.
  8. What was the criteria for selection of six concentration levels for each analyte (ranging from 5 to 500 μg/L). Is response and levels for all 33 pesticides same? If not, selection of the same linearity range for all 33 pesticides may not be valid. 
  9. Table 3: In general convention Q1 is precursor, however, it appears the author mentioned Q1 as fragment ions. What are the precursors?
  10. Too many tables which can be moved to supplementary information or can be summarized in concise figures
  11. Table 4,5 and 6 provides less information that can be moved to SI or summarized in a better way.
  12. Please provide more comparison with background/ literature of earlier reported methods and conclusion with this study. 

Author Response

Dear Reviewer,

Thank you for the opportunity to revise the paper. We found all the opinions very pertinent and helpful, and they helped to improve the manuscript. In the following you will find our answers to each of the comments.

Kind Regards,

Salem Hayar

Comments and Suggestions for Authors

The study highlights the important concern about deregulated use of pesticides in Labanese marker of vine leave. The authors have validated the method and analyzed 17 marketed vine leaves in 3 forms. The aim of the study is to validate the method, assess the current level of pesticide V. Vinifera leaves on the Lebanese market.

However, the study raises following concerns about method and conclusion.

  1. How the major observation about more pesticides found in dried leave is validated. For example; author should test that sample preparation of brine and stuffed samples does not eliminate any pesticides particularly, the washing step. It might be the case that pesticides are eliminated during sample preparation and washing which gives the impression that dried leaves have more pesticides. 

The analyte stability during the washing procedure of the stuffed vine leaves market samples was also checked as part of the method validation. Previous studies showed that the washing procedure with deionized water did not significantly affect the residues in stuffed vine leaves, even for the water soluble compounds such as acetamiprid, dimethoate and metalaxyl (unpublished data).

The results presented in this study are part of a huge data analysis lasted more than 5 years which took into account the monitoring of the dissipation rate of more than 30 pesticide residues in vine leaves during 4 months of conservation, for the 3 conservation methods (dry, brine and stuffed). In these unpublished results we have succeeded in assessing a half-life for each molecule. This work had shown that the difference in pesticide residues between the 3 conservation methods is not due to the washing of the leaves. In addition, these results were confirmed by a preliminary study on the pesticide residue content of brine and the washing water. These data will be presented through another scientific paper once the validation of the method is published.

The text was amended (line 152-156).

We add below an example of two dissipation patterns of the pesticide residues for dimethoate and chlorpyrifos during storage time for the three conservation methods (Dry, Brine and Stuffed).

  1. The stability of pesticides during storage and preparation is not checked. 

As described previously, our aim in this study was not to assess the stability of pesticides during storage or preparation, but to make an inventory of the levels of pesticide residues that may occur in the 3 methods of preservation to answer the question of which of these preservation methods is safer and healthier for the consumer.

  1. The above two points raises some concern about the comparative data presented for 3 types of samples. 

The explanation was given in answers 1 and 2.

  1. Is the LC-MS method new or adapted from literature. What is the improvement or advantage of the LC-MS method over previous reports. 

The results presented in this study are validated according to the QuEChERS method using an LCMSMS and acetonitrile as solvent extraction, and adapted to the local conditions of the laboratory, including the Mass spectrometric conditions. Particularly the LC-MS/MS system was equipped with a pre-configured iMethod™ Application (AB Sciex, USA) and associated libraries which helped the analytical determination. The limits of quantification were lower than the corresponding default EU-MRLs for vine leaves, rendering the method acceptable for checking compliance to MRLs. The analytes reported in this study are different from those reported in previous literature.

  1. Author didn’t explain the observed pesticide levels and their possible causes w.r.t. sample sources. 

The samples were collected from the market and the label did not indicate the origin, therefore the traceability, and any explanation thereof, was compromised.

  1. Method section need more details about sample collection, properties, selection criteria, LC-MS method optimization, selection of MS/MS transitions, LC column (C18 or C8)

Section 6 was amended in the text at lines 326-333.

Chromatographic separation was performed with a reverse-phase analytical C18 column of 150 x 2 mm and 2.5 μm particle size, Synergi (Phenomenex, USA) equipped with a guard-column. The mobile phase consisted of a water-methanol solvent containing each one 5mM ammonium acetate. The gradient program was as follows: 2% B to 100% of B over 12 min, held at 100% B until 20 min then decreased to 2% B at 25.01 min. The total run time was 35 min with a flow rate of 0.4 mL min-1. The injection volume was 5 μL and the column was maintained at 25°C.

  1. Identification of pesticides is not clear. Please provide some examples of MS/MS spectra from standard and sample.

An example of TIC is presented in Figure 3 to show the identification of a positive sample and a matrix matched standard.

  1. What was the criteria for selection of six concentration levels for each analyte (ranging from 5 to 500 μg/L). Is response and levels for all 33 pesticides same? If not, selection of the same linearity range for all 33 pesticides may not be valid.

The criteria used for choosing multiple calibration levels is given by the SANTE guideline 12682 /2019 (point C17). Similar criteria are in ISO 11095. The linearity was checked and demonstrated that all analytes had a linear calibration range. The linearity for the 33 pesticides were all greater than 0.99%.

  1. Table 3: In general convention Q1 is precursor, however, it appears the author mentioned Q1 as fragment ions. What are the precursors?

The headings in Table 3 were confusing, Table 3 was amended.

  1. Too many tables which can be moved to supplementary information or can be summarized in concise figures

Table 1 and Table 3 were moved to the Supplementary materials.

  1. Table 4,5 and 6 provides less information that can be moved to SI or summarized in a better way.

The 3 tables were merged into one.

  1. Please provide more comparison with background/ literature of earlier reported methods and conclusion with this study. 

The text and the references were amended (111-124).

In recent years a very limited number of compounds have been studied in vine leaves, for example, trifloxystrobin, tebuconazole [35], fipronil [36], imidacloprid [37], azoxystrobin, fenhexamid and lufenuron [38]. Only two studies are available for the multi residue validation in grape vine leaves. Authors reported a gas chromatography coupled to tandem mass spectrometry GC-MS/MS method where the sample preparation involved a modified and miniaturized SweEt/QuEChERS method, which uses acidified ethyl acetate for extraction (SweEt) and cleanup using a modified QuEChERS procedure. This method was only tested for 59 GC-amenable pesticides [39]. Other authors developed a comprehensive analytical method for the identification and quantification of a broad range of pesticides and plant growth regulators in vine leaves matrix using an optimized QuEChERS-based extraction protocol and determination by GC-MS/MS and liquid chromatography coupled to tandem mass spectrometry LC-MS/MS. Both methods were validated showing satisfactory recoveries, accuracy, precision in compliance with CODEX and SANTE’s method performance criteria

[40]. Jyot, G., Arora, P.K., Sahoo, S.K., Singh, B., Battu, R.S. Persistence of trifloxystrobin and tebuconazole on grape leaves, grape berries and soil, Bull. Environ. Contam. Toxicol. 2010, 84, 305-310.

  1. Mohapatra, S., Deepa, M., Jagdish, G.K., Rashmi, N., Kumar, S., Prakash, G.S. Fate of fipronil and its metabolites in/on grape leaves, berries and soil under semi-arid tropical climatic conditions, Bull. Environ. Contam. Toxicol. 2015, 84, 587-591.
  2. Arora, P.K., Jyot, G., Singh, B., Battu, R.S., Singh, B., Aulakh, P.S. Persistence of imidacloprid on grape leaves, grape berries and soil, Bull. Environ. Contam. Toxicol. 2009, 82, 239-42.
  3. Manal, R., Montasser, H., Mahmoud, A. Chromatographic determination of azoxystrobin, fenhexamid and lufenuron residues in grapevine, Alexandria Sci. Exchange J. 2009, 30, 38-44.

Submission Date

20 January 2021

Date of this review

28 Jan 2021 10:02:59

Reviewer 2 Report

Dear the Editor,

Hayar S et al reported the summary of validation for pesticides in wine products by LC-MS/MS combined with QuEChERS method. Overall, the recovery of pesticides spiked into the sample reached between 75 to 104%, indicating that the these analytes were extracted efficiently. Then, these authors further applied their assay procedure for Lebanese wine products and further discussed effect of three different preservation procedures, such as dry, brine, and stuffed technique. This manuscript seemed to be well-organized and delivers fruitful information to the audience of the Molecule journal.

Author Response

Dear Reviewer,

Thank you for the opportunity to revise the paper.

Kind Regards,

Salem Hayar

Comments and Suggestions for Authors

Dear the Editor,

Hayar S et al reported the summary of validation for pesticides in wine products by LC-MS/MS combined with QuEChERS method. Overall, the recovery of pesticides spiked into the sample reached between 75 to 104%, indicating that the these analytes were extracted efficiently. Then, these authors further applied their assay procedure for Lebanese wine products and further discussed effect of three different preservation procedures, such as dry, brine, and stuffed technique. This manuscript seemed to be well-organized and delivers fruitful information to the audience of the Molecule journal.

Submission Date

20 January 2021

Date of this review

28 Jan 2021 12:33:23

Reviewer 3 Report

Introduction line 105-110 - The author should clarify the information on the EUMRLs, the crop “grape leaves and similar species” exist in the EU MRL database and the MRLs are set. I Propose to author to revise the text according to the EU Reg 396/2005 and its amending. The text in the line 105-110 appear in contrast with the tables 4, 5, and 6 where are reported MRLs.

Materials and method 3.1.1 market samples – the storage condition of the market samples is not appropriate for the analysis of residue. According to the document SANTE/12682/2019 B6 “Fresh products may be stored in the refrigerator but tipically no longer than 5 days…” the author give a justification on the storage condition.

3.5 LC-MS/MS analysis – The gradient should be described.

3.6 Identification… - Line 346 update the SANTE document also in all text with SANTE/12682/2019

A typical chromatogram of samples should be given

Author Response

Dear Reviewer,

Thank you for the opportunity to revise the paper. We found all the opinions very pertinent and helpful, and they helped to improve the manuscript. In the following you will find our answers to each of the comments.

Kind Regards,

Salem Hayar

Comments and Suggestions for Authors

Introduction line 105-110 - The author should clarify the information on the EUMRLs, the crop “grape leaves and similar species” exist in the EU MRL database and the MRLs are set. I Propose to author to revise the text according to the EU Reg 396/2005 and its amending. The text in the line 105-110 appear in contrast with the tables 4, 5, and 6 where are reported MRLs.

The EU database indicates, through an asterix, that the MRLs for vine leaves have to be interpreted as the default MRL = 0.01 mg/kg, corresponding to the lower limit of analytical determination. The MRLs in the tables are those corresponding to the EU MRLs. The tables are amended.

Materials and method 3.1.1 market samples – the storage condition of the market samples is not appropriate for the analysis of residue. According to the document SANTE/12682/2019 B6 “Fresh products may be stored in the refrigerator but tipically no longer than 5 days…” the author give a justification on the storage condition.

The market samples were stored intact, in their original packaging, and stored in the dark at room temperature (22°C) until analysis. The market samples were not fresh samples. The text was amended. (Line 275-276).

3.5 LC-MS/MS analysis – The gradient should be described.

The text was amended. (line 326-333)

Chromatographic separation was performed with a reverse-phase analytical C18 column of 150 x 2 mm and 2.5 μm particle size, Synergi (Phenomenex, USA) equipped with a guard-column. The mobile phase consisted of a water-methanol solvent containing each one 5mM ammonium acetate. The gradient program was as follows: 2% B to 100% of B over 12 min, held at 100% B until 20 min then decreased to 2% B at 25.01 min. The total run time was 35 min with a flow rate of 0.4 mL min-1. The injection volume was 5 μL and the column was maintained at 25°C.

3.6 Identification… - Line 346 update the SANTE document also in all text with SANTE/12682/2019

The text was amended according to the latest version of SANTE.

3.7. A typical chromatogram of samples should be given.

A figure was included. (figure 3).

Submission Date

20 January 2021

Date of this review

29 Jan 2021 09:00:47

REVIEWER 4

Open Review

English language and style

( ) Extensive editing of English language and style required
( ) Moderate English changes required
( ) English language and style are fine/minor spell check required
(x) I don't feel qualified to judge about the English language and style

Yes

Can be improved

Must be improved

Not applicable

Does the introduction provide sufficient background and include all relevant references?

(x)

( )

( )

( )

Is the research design appropriate?

(x)

( )

( )

( )

Are the methods adequately described?

(x)

( )

( )

( )

Are the results clearly presented?

(x)

( )

( )

( )

Are the conclusions supported by the results?

(x)

( )

( )

( )

Comments and Suggestions for Authors

The manuscript entitled “Validation of a rapid multiresidue method for the determination of pesticide residues in vine leaves. Comparison of the results according to the different conservation methods” validated a method for pesticide residue analysis in vine leaves by liquid chromatography coupled to tandem mass spectrometry and compared the pesticide residue content in vine leaves processed using three different preservation methods. The topic of the manuscript is of interest for the readers of Molecules and gives new information, and for these reasons should be accepted after minor revisions.

Comments:

All abbreviations should be reported when used for the first time.

The text was amended.

The authors should pay attention to significant numbers.

The text was amended.

Table 3. Please added the linearity achieved for each pesticides.

The table was amended.

An exemplary chromatogram of analysed compounds should be provided.

A figure was included.

Submission Date

20 January 2021

Date of this review

23 Jan 2021 17:32:30

Reviewer 4 Report

The manuscript entitled “Validation of a rapid multiresidue method for the determination of pesticide residues in vine leaves. Comparison of the results according to the different conservation methods” validated a method for pesticide residue analysis in vine leaves by liquid chromatography coupled to tandem mass spectrometry and compared the pesticide residue content in vine leaves processed using three different preservation methods. The topic of the manuscript is of interest for the readers of Molecules and gives new information, and for these reasons should be accepted after minor revisions.

Comments:

All abbreviations should be reported when used for the first time.

The authors should pay attention to significant numbers.

Table 3. Please added the linearity achieved for each pesticides.

An exemplary chromatogram of analysed compounds should be provided.

Author Response

Dear Reviewer,

Thank you for the opportunity to revise the paper. We found all the opinions very pertinent and helpful, and they helped to improve the manuscript. In the following you will find our answers to each of the comments.

Kind Regards,

Salem Hayar

Comments and Suggestions for Authors

The manuscript entitled “Validation of a rapid multiresidue method for the determination of pesticide residues in vine leaves. Comparison of the results according to the different conservation methods” validated a method for pesticide residue analysis in vine leaves by liquid chromatography coupled to tandem mass spectrometry and compared the pesticide residue content in vine leaves processed using three different preservation methods. The topic of the manuscript is of interest for the readers of Molecules and gives new information, and for these reasons should be accepted after minor revisions.

Comments:

All abbreviations should be reported when used for the first time.

The text was amended.

The authors should pay attention to significant numbers.

The text was amended.

Table 3. Please added the linearity achieved for each pesticides.

The test was amended.

An exemplary chromatogram of analysed compounds should be provided.

A figure was included.

Submission Date

20 January 2021

Date of this review

23 Jan 2021 17:32:30

Round 2

Reviewer 1 Report

Please make necessary changes in the manuscript according to the comments. Comments about stability, comparative methods, and validation criteria including limitations of the study (origin of the samples) are only answered as responses to reviewer comments. Please make necessary changes in your manuscript accordingly. 

Author Response

  • Concerning the stability:

The text was amended (line 161-167).

  • Concerning the comparative methods:

The text was amended and clarification was mentioned (line 111-124+127+345-346).

  • Concerning the validation criteria:

The text was amended (line 151-153).

  • Concerning the origin of the samples:

Clarification was mentioned (line 264-269).